# Social isolation-induced transcriptomic changes in mouse hippocampus impact the synapse and show convergence with human genetic risk for neurodevelopmental phenotypes

Aodán Laighneach[1], John P. Kelly[2], Lieve Desbonnet[2], Laurena Holleran[1], Daniel M. Kerr[2], Declan McKernan[2], Gary Donohoe[1], Derek W. Morris[1]*

1 Centre for Neuroimaging, Cognition and Genomics (NICOG), School of Biological and Chemical Sciences and School of Psychology, University of Galway, Galway, Ireland, 2 Discipline of Pharmacology and Therapeutics, School of Medicine, University of Galway, Galway, Ireland

* derek.morris@universityofgalway.ie

**Data Availability Statement:** All relevant data are within the paper and its Supporting information

## Abstract

Early life stress (ELS) can impact brain development and is a risk factor for neurodevelopmental disorders such as schizophrenia. Post-weaning social isolation (SI) is used to model ELS in animals, using isolation stress to disrupt a normal developmental trajectory. We aimed to investigate how SI affects the expression of genes in mouse hippocampus and to investigate how these changes related to the genetic basis of neurodevelopmental phenotypes. BL/6J mice were exposed to post-weaning SI (PD21-25) or treated as group-housed controls (n = 7–8 per group). RNA sequencing was performed on tissue samples from the hippocampus of adult male and female mice. Four hundred and 1,215 differentially-expressed genes (DEGs) at a false discovery rate of < 0.05 were detected between SI and control samples for males and females respectively. DEGS for both males and females were significantly overrepresented in gene ontologies related to synaptic structure and function, especially the post-synapse. DEGs were enriched for common variant (SNP) heritability in humans that contributes to risk of neuropsychiatric disorders (schizophrenia, bipolar disorder) and to cognitive function. DEGs were also enriched for genes harbouring rare *de novo* variants that contribute to autism spectrum disorder and other developmental disorders. Finally, cell type analysis revealed populations of hippocampal astrocytes that were enriched for DEGs, indicating effects in these cell types as well as neurons. Overall, these data suggest a convergence between genes dysregulated by the SI stressor in the mouse and genes associated with neurodevelopmental disorders and cognitive phenotypes in humans.

## Introduction

Early life stress (ELS) includes childhood exposure to a range of adversities, and is associated with increased risk for neurodevelopmental disorders [1]. One such stressor is social isolation

files. Raw RNA-seq data is available under GEO accession number GSE246551.

**Funding:** This work was funded by grants from the European Research Council (ERC-2015- STG-677467 to GD; https://erc.europa.eu/), Science Foundation Ireland (SFI- 16/ERCS/3787 to GD; https://www.sfi.ie/) and the Irish Research Council (GOIPG/2019/1932 to AL; https://research.ie/). The funders had no role in study design, data collection and analysis, decision to publish, or preparation of the manuscript.

**Competing interests:** The authors have declared that no competing interests exist.

(SI), a situation in which an individual is deprived of typical and expected social interaction. These interactions are fundamental to normal social development [2] and exposure to SI during vulnerable periods of neurodevelopment can impact on both the behaviour and neurobiology of those affected [3, 4]. The detriment of SI-induced changes is clear, with evidence that exposure contributes to increased risk of anxiety and depression [5, 6], schizophrenia [7] and cognitive decline [8].

The use of rodent models of SI to investigate behaviour is common. Mice exposed to post-weaning SI tend to experience increased anxiety [9–16], increased depressive-like behaviour [11, 13], decreased cognitive ability [13, 14, 17–19], increased aggression [12, 20] and decreased sociability [10, 21] when compared to group-housed controls. Some of the neurobiology relevant to SI-induced behaviour in rodents has been mapped [22]. Targeted analyses have implicated neuronal growth factors such as BDNF [13, 16, 17, 20], hormones such as oxytocin [23], inflammatory mediators such as cytokines [24–26] and the function of most main neurotransmitter systems [22] as being involved in behaviours found in SI-exposed rodents. Rodents subjected to SI display dysregulation of genes crucial to the function of glutamatergic [14, 27, 28], GABAergic [28–30], dopaminergic [31] neurotransmission, all of which are systems thought to have a part to play in psychiatric disorders [32].

To our knowledge, no mouse studies of SI have included full transcriptome gene expression analysis of the molecular changes caused by SI in the hippocampus. Previously, transcriptomic analysis of the basolateral amygdala in socially-isolated mice identified genes associated with aggressive behaviour and found evidence of upregulated ion channel function, while genes related to limbic system development and cognition were downregulated. Furthermore, in the ventral tegmental area, genes related to neuropeptide signalling were downregulated and genes related to synaptic signalling were upregulated [33]. Other transcriptomic work using microarray in rat cortex [29] also found dysregulation of genes related to synaptic structure and function, particularly in relation to inhibitory GABAergic synapses.

Previous work by our group [10] showed that mice exposed to post-weaning SI exhibited a significant increase in anxiety related behaviours (males and females) and decreased sociability (females) compared to animals that were group-housed. We now extend our previous investigation by performing RNA sequencing (RNA-seq) to identify differentially expressed genes (DEGs) in the brains of the same animals–specifically in the hippocampus, a region that regulates stress response and emotion [34, 35]. Transcriptomic changes were found to be induced in both male and female hippocampus by SI. These differentially expressed genes (DEGs) encoded protein involved in synaptic structure and function. We further considered how these transcriptomic changes converge with the biological basis of human psychiatric disorders and behaviours by investigating if the DEGs were enriched for common heritability contributing to neurodevelopmental phenotypes and enriched for genes harbouring rare *de novo* variants contributing to neurodevelopmental disorders. Finally, we explored which specific cell types were enriched for DEGs induced by SI.

## Materials and methods

### Ethics statement

All procedures received ethical approval from the local Animal Care Research Ethics Committee (ACREC) and the Health Products Regulatory Authority in Ireland (licence number AE19125/P083).

## Social isolation procedure

Tissues used in this analysis were obtained directly from the animals used in our previous study [10], consisting of post-weaning (P21-25) adult male and female C57 BL/6J mice (Charles River Laboratories, UK) assigned to either group-housed cages (3–4 animals per cage) or single housed cages (SI; n = 7–8 per group) until animals were sacrificed by brief exposure to $CO_2$ (90s) followed by immediate decapitation after a 60 day period. Whole brain was dissected on ice and hippocampus was snap-frozen in dry-ice before being stored at −80 ˚C.

## Sample preparation

Total RNA was extracted from frozen hippocampus using the Absolutely RNA Miniprep kit (Agilent; product code: #400800) following appropriate standard procedure for quantity of tissue. Once purity and integrity (RIN > 6) was confirmed, isolated RNA-seq was performed on the Illumina NovaSeq (paired-end; 2x150 bp reads) sequencing platform producing a minimum of > 20M reads/sample (Genewiz Germany GmbH).

## Differential expression analysis

Raw data was received through SFTP in FASTQC format. *Trimmomatic* v0.39 [36] was used to remove low quality and adapter sequences from the paired-end reads (LEADING:3, TRAILING:3, MINLEN:36). *Salmon* v1.8.0 [37] was used to quasi-map and quantify reads. The *DEseq2* v1.24.0 [38] R package was used to test genes for differential expression. The *sva* v3.32.1 *svaseq* [39] function was used to detect batch effects in individual groups. Significant surrogate variable (SVs) were introduced into the differential expression model to control for technical batch effects. DEGs were defined at a false discovery rate (FDR) of < 0.05. Fold changes were shrunk using *apeglm* v1.6.0 [40]. Genes were converted to human orthologues using biomaRt v2.40.5 [41] where necessary.

## Gene ontology analysis

Gene ontology (GO) analysis was done using ConsensusPathDB (http://cpdb.molgen.mpg.de/) [42] over-representation analysis. Ontologies with GO term levels 2–5 were tested and ontologies with a FDR-corrected p-value < 0.05 were considered significantly enriched. In order to limit GO analysis to ontologies relevant to the synapse, sets of DEGs also tested for enrichment using SynGO (https://www.syngoportal.org/) [43], an expert-curated resource for synaptic GO analysis. Ontologies with an FDR corrected p-value < 0.05 were considered significantly enriched.

## Testing genes for enrichment of common genetic risk variants associated with neurodevelopmental phenotypes

Data on common variants (SNPs) associated with human phenotypes were accessed in the form of genome-wide association study (GWAS) summary stats for a range of phenotypes relevant to neurodevelopment and psychiatric disorders including schizophrenia (SCZ; GWAS based on 67,390 cases and 94,015 controls) [44], intelligence (IQ; 269,867 individuals) [45], educational attainment (EA; 766,345 individuals) [46], bipolar disorder (BPD; 41,917 cases and 371,549 controls) [47], major depressive disorder (MDD; 246,636 cases and 561,190 controls) [48] and anxiety-tension (Anx-Ten; 270,059 individuals) [49]. As control phenotypes, GWAS data for three brain-related disorders: Attention deficit/hyperactivity disorder (ADHD; 20,183 cases and 35,191 controls) [50], Alzheimer's disease (AlzD; 71,880 cases and

383,378 controls) [51], stroke (40,585 cases and 406,111 controls) [52] and two non-brain related disorders: type-2 diabetes (T2D; 74,124 cases and 824,006 controls) [53] and coronary artery disease (CAD; 22,233 cases and 64,762 controls) [54] were used. Stratified LD Score regression (sLDSC) was used to investigate if gene-sets were significantly enriched for SNP heritability contributing to the test and control phenotypes [55, 56]. Gene start and stop coordinates of each DEG on GRCh37 were found using biomaRt v2.40.5 [41]. Annotation files were generated for each chromosome in each set of DEGs using 1000 genomes European cohort SNPs and a window size of 100kb extended to coordinates [57]. LD scores were estimated within a 1cM window using 1000 Genomes Phase 3 European reference panel. Heritability was stratified in a joint analysis between 53 previous function genomic annotations [56] and each set of DEGs. Only SNPs from HapMap Project phase 3 SNPs with a MAF > 0.05 were considered in this analysis.

For phenotypes with significantly enriched heritability from the LDSC analysis, MAGMA [58] was used to test individual genes for association. First, we annotated the SNP data to genes using the build 37 gene locations (https://ctg.cncr.nl/software/MAGMA/aux_files/NCBI37.3.zip) and 1000 Genomes European Panel reference (https://ctg.cncr.nl/software/MAGMA/ref_data/g1000_eur.zip) files, the latter which MAGMA uses to account for linkage disequilibrium (LD) between SNPs. Second, MAGMA generated p-values for individual gene reflecting their level of association with the test and control phenotypes. Genes with a Bonferroni-corrected p-value of $< 0.05$ (correcting for the number of genes tested) were considered genome-wide significant for association with each phenotype.

### Testing genes for enrichment of *de novo* mutations reported in neurodevelopmental disorders

Rare *de novo* mutations (DNMs) contributing to a phenotype can be detected using exome sequencing of trios including an affected proband and their biological parents. To test if DEGs were enriched for DNMs that contribute to neurodevelopmental disorders, we analysed DNMs reported in studies of autism spectrum disorder (ASD), intellectual disability (ID), SCZ and developmental disorders (DD). The functional class and gene location of DNMs identified in patients with ASD (n = 6,430), ID (n = 192) and in unaffected siblings (n = 1,995) and controls (n = 54) based on exome sequencing of trios were sourced from [59] and [60]. Genes harbouring DNMs reported in affected SCZ trios (n = 3,394) were taken from [61] and [62]. DNMs identified in developmental disorders (DD) were sourced from [63]. Mutations used for DD were subject to additional filtering based on posterior probability of *de novo* mutations, as described in [63]. To account for underlying mutational burden associated with the test phenotypes, results were then subject to a competitive test against background *de novo* mutation rate using a two-sample Poisson rate ratio test. Results with a Bonferroni-adjusted p-value of $< 0.05$ (correcting for the number of mutational classes and disorders tested) were considered significantly enriched.

### Cell type enrichment

Data from single cell RNA-seq (scRNA-seq) of the mouse brain (565 cell types) [64] was used to test if different cell types were enriched for DEGs. Analysis was performed using the expression-weighted cell type enrichment (EWCE) R package [65], which investigated whether the cell types were significantly enriched for a gene-set when weighted by gene expression. Cell types were considered significantly enriched at a Bonferroni-corrected p-value of $< 0.05$ (correcting for the number of cell types tested).

**Table 1. Summary of DEGs induced by SI in hippocampus.**

| Set | DEGs at FDR < 0.05 | Downregulated | Upregulated |
|---|---|---|---|
| Female | 1215 | 810 (67.7%) | 405 (33.3%) |
| Male | 400 | 259 (64.7%) | 141 (35.3%) |

## Results

### Differential gene expression

Table 1 summarises the numbers of DEGs induced by SI detected in male and female hippocampal tissue at FDR < 0.05. In total, 1,215 significant DEGs were identified in females and 400 in males. In both sexes, approximately twice as many DEGs were found to be downregulated than upregulated. Among the downregulated DEGs, 98 were common between males and females. Of the upregulated DEGs, 30 were common between males and females. Expression log 2 fold change (Log2FC) was highly consistent among DEGs shared between male and female samples, producing a $R^2$ correlation coefficient of 0.88 (p-value < 2.2e-16). Full differential expression results can be found in S1 Table.

### GO and SynGO analysis

GO analysis was performed in order to gain insight into the functional roles of DEGs. ConsensusPathDB was used to identify GO biological processes (BP), cellular compartments (CC) and molecular functions (MF) terms enriched for DEGs. A total of 738 GO terms were found to be enriched for the female DEGs and 306 GO terms were enriched for the male DEGs at FDR < 0.05. (S2 Table). Of these enriched terms, 72 (9.8%) of the female and 36 (11.8%) of the male contained any of "neuro", "synap", "dendr" or "axon", indicating their relation to having neuronal, axonal or synaptic structure or function. Female DEGs were highly enriched for a number of neurodevelopmental GO terms. The most enriched level 4 and level 5 terms were *nervous system development* (GO:0007399, FDR-adjusted p-value = q-value = 2.76E-16) *neurogenesis* (GO:0022008, q-value = 9.02E-16), *neuron development* (GO:0048666, q-value = 1.26E-15), *generation of neurons* (GO:0048699, q-value = 1.57E-15), *neuron projection development* (GO:0031175, q-value = 1.57E-15) and *neuron differentiation* (GO:0030182, q-value = 2.57E-15). The significantly enriched GO terms for the male DEGs included *nervous system development* (GO:0007399, q-value = 6.79E-6), *neuron part* (GO:0097458, q-value = 9.82E-5), *central nervous system development* (GO:0007417, q-value = 3.31e-4) and *brain development* (GO:0007420, q-value = 4.61e-4).

Following up these neuronal GO term enrichments, additional GO analysis was performed using SynGO [43] to gain insight into the synaptic CCs and BPs enriched for DEGs (Fig 1; S3 Table). All enriched SynGO analysis terms were related to presynaptic and postsynaptic structure (CC) and function (BP) and provide greater focus on the precise neurobiological changes brought about by SI. The female DEG set showed significant enrichments in 39 SynGO terms. Both *postsynapse* (GO:0098794, q-value = 5.14e-17) and *presynapse* (GO:0098793, q-value = 3.15e-4) CCs were significantly enriched as well as a number of specific postsynaptic ontologies including *postsynaptic density* (GO:0014069, q-value = 2.25e-8), *postsynaptic specialisation* (GO:0099572, q-value = 1.56e-7), *translation at postsynapse* (GO:0140242, q-value = 5.76e-4) and *postsynaptic organisiation* (GO:0099173, q-value = 2.19e-3). The male DEGs showed significant enrichment in the *postsynapse* (GO:0098794, q-value = 3.67e-3) but not *presynapse* CCs. Similar to that of the females, male DEGs were also significantly enriched

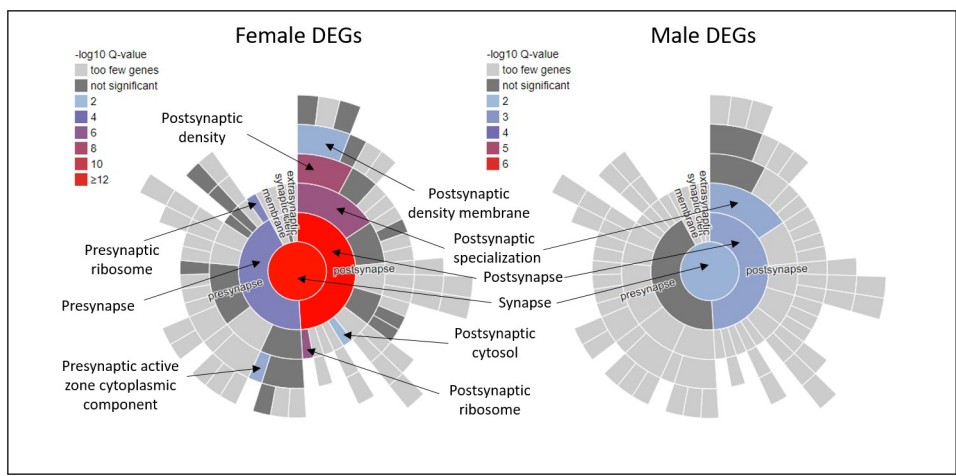

**Fig 1. SynGO cellular compartment (CC) enrichments of female and male DEG sets.** Annotated sunburst of enriched SynGO synaptic CC ontologies. Enrichments were detected for both pre and post-synaptic terms, however most significantly-enriched ontologies are post-synaptic. DEGs from females showed more enrichments in SynGO terms than DEGs from males.

for processes of postsynaptic specialisation (GO:0099572, q-value = 0.003667) and *postsynaptic density* (GO:0014069, q-value = 0.012).

## Enrichment for genes contributing to human neurodevelopmental phenotypes

To assess whether genes dysregulated in mouse hippocampal tissue by SI harboured genetic risk for human neurodevelopmental phenotypes, enrichments for common heritability (SNPs) and genes harbouring rare DNMs contributing to neurodevelopmental disorders and related phenotypes were investigated. sLDSC [55, 56] was used to test common SNPs. The 1,215 female DEGs accounted for 9.5% of the total SNPs in the analysis, while the smaller male set of 400 DEGs accounted for 3.5% of total SNPs. Results can been seen in Table 2, showing percentage of heritability (% $h^2$) and enrichment p-value in SNP-based heritability for SCZ, EA, IQ and BPD for the female DEG set and enrichments in SCZ, IQ and BPD for the male DEG set. Of the five control phenotypes tested, just one (coronary artery disease) showed a marginal enrichment for the female DEG set. Full results from the sLDSC analysis can be found in S4 Table.

Given there was evidence of convergence of SI-induced DEGs and genes contributing to risk of neurodevelopmental phenotypes, we investigated which genes may be driving this connection. For the test phenotypes with significant enrichments in LDSC (SCZ, EA, IQ and BPD), MAGMA [58] was used to test which genes had genome-wide significant associations.

**Table 2. Enrichment of SNP-based heritability for test phenotypes.**

| DEG Set | % of Total SNPs | SCZ | | IQ | | EA | | MDD | | BPD | | Anx-Ten | |
|---|---|---|---|---|---|---|---|---|---|---|---|---|---|
| | | % $h^2$ | P* | % $h^2$ | P | % $h^2$ | P | % $h^2$ | P | % $h^2$ | P | % $h^2$ | P |
| Female | 9.5% | 13.3% | **2.27E-05** | 12.9% | **0.000109** | 13.3% | **2.60E-07** | 11.7% | 0.0472 | 14.2% | **0.000388** | 12.3% | 0.0422 |
| Male | 3.5% | 6.2% | **7.08E-05** | 5.0% | 0.00276 | 4.5% | 0.0132 | 4.6% | 0.0434 | 7.2% | **2.51E-05** | 4.1% | 0.4202 |

Results in **bold** survive Bonferroni correction for multiple testing (including control phenotypes)

**Table 3. Genes found to be common between DEG analysis (same direction of effect in females and males) and *MAGMA* gene-based association analysis.**

| Symbol | Gene Name | Associated Phenotypes | Direction in Females | Direction in Males |
|--------|-----------|----------------------|---------------------|-------------------|
| RIMS1 | Regulating Synaptic Membrane Exocytosis 1 | SCZ, BPD | Downregulated, Log2FC = -0.22 | Downregulated, Log2FC = -0.32 |
| ZNF365 | Zinc Finger Protein 365 | SCZ, BPD | Downregulated, Log2FC = -0.21 | Downregulated, Log2FC = -0.32 |
| AGAP1 | ArfGAP With GTPase Domain, Ankyrin Repeat And PH Domain | IQ, EA | Downregulated, Log2FC = -0.21 | Downregulated, Log2FC = -0.25 |
| TTBK1 | Tau Tubulin Kinase 1 | SCZ | Downregulated, Log2FC = -0.14 | Downregulated, Log2FC = -0.31 |
| PPP1R16B | Protein Phosphatase 1 Regulatory Subunit 16B | SCZ | Downregulated, Log2FC = -0.18 | Downregulated, Log2FC = -0.31 |
| SPTBN2 | Spectrin Beta, Non-Erythrocytic 2 | BPD | Downregulated, Log2FC = -0.21 | Downregulated, Log2FC = -0.32 |
| SHANK2 | SH3 And Multiple Ankyrin Repeat Domains 2 | BPD | Downregulated, Log2FC = -0.14 | Downregulated, Log2FC = -0.27 |
| UTRN | Utrophin | EA | Downregulated, Log2FC = -0.25 | Downregulated, Log2FC = -0.24 |
| PHF2 | PHD Finger Protein 2 | IQ | Downregulated, Log2FC = -0.12 | Downregulated, Log2FC = -0.18 |
| ZBTB4 | Zinc Finger And BTB Domain Containing 4 | EA | Downregulated, Log2FC = -0.13 | Downregulated, Log2FC = -0.33 |

Genes were considered significantly enriched at a Bonferroni-corrected p-value of < 0.05. A total of 619 genes were genome-wide significant for SCZ, 160 for BPD, 377 for IQ and 689 for EA. Full MAGMA gene-based analysis results can be found in S5 Table. These lists were further restricted to DEGs that were common between male and female hippocampus and with consistent direction of expression change. A total of ten genes, shown in Table 3 below, matched these criteria and were significantly associated with one or more of SCZ, BPD, IQ or EA. Three of the 10 highlighted genes were genome-wide significant for two phenotypes, while 7 had a single significant gene-phenotype association from MAGMA. At least one gene was associated with SCZ, BPD, IQ or EA. All ten genes were downregulated, with male hippocampus consistently showing greater expression change.

Using gene-level data from these studies, the R package *denovolyzeR* [66] was used to test gene-sets for enrichment for genes harbouring rare DNMs contributing to SCZ, ASD, ID and DD. Categories of synonymous (syn), missense (mis) and loss-of-function (lof) mutations were considered for analysis. Following a competitive analyses, gene-sets were considered enriched at a Bonferroni-corrected p-value of <0.05. Table 4 shows enriched categories of

**Table 4. Enrichments for genes harbouring rare *de novo* mutations in male and female gene sets.**

| Set | DEGs | SCZ n = 3394 Trios | ASD n = 6430 Trios | ID n = 192 Trios | DD n = 4293 Trios |
|-----|------|---------------------|---------------------|-------------------|--------------------|
| Female Hippocampus | 1215 | ns | ns | ns | **mis (p = 5.20e-6)** |
| Male Hippocampus | 400 | ns | **lof (p = 2.60e-5)** | ns | **lof (p = 8.23e-8) mis (p = 8.05e-5)** |

SCZ = Schizophrenia; ASD = Autism; ID = Intellectual Disability; DD = Developmental Disorder

ns = non-significant; lof = loss of function mutations; mis = missense mutations; Results in **bold** survive Bonferroni correction for multiple testing

variants in each gene-set. Female DEGs showed strong enrichment for genes containing missense variants contributing to DD. Male DEGs were also enriched for missense variants contributing to DD as well as being enriched for genes containing loss-of-function mutations contributing to DD and ASD. As a control, gene-sets were not enriched for synonymous variants in any of the disorders. No enrichments were seen in trios containing unaffected siblings or controls. Complete results from rare variant analysis can be found in S6 Table.

### Cell-type enrichment analysis

Cell types enriched for male and female DEGs were tested using EWCE [65]. Both sets were tested in scRNA-seq gene expression data from mouse brain [64]. Although this dataset contains expression data on 565 cell types, enrichments from this analysis was restricted to cell types from the hippocampus (n = 104). Cell types were considered enriched at a Bonferroni-corrected p-value of $< 0.05$. The primary cell type found to be enrichment in female DEGs was of glial origin. Of the cell types in the mouse brain data [64] three hippocampal cell types, HC_7–1 ($p < 0.00001$), HC_7–2 ($p < 0.00001$) and HC_7–3 ($p < 0.00001$) were enriched; all three were populations of hippocampal astrocytes. No hippocampal cell types were considered significantly enriched after multiple testing correction in the male DEG set. Full cell type enrichment results found in S7 Table.

## Discussion

This study builds on our previous behavioural work [10] using post-weaning SI mice and here investigates the molecular consequences of the environmental stressor on the mouse brain. The use of RNA-seq to investigate the transcriptomic changes in mouse hippocampus caused by SI presents a novel insight into the underlying molecular underpinnings of altered behaviour. Furthermore, we used data from studies of rare and common risk variants for neurodevelopmental phenotypes to investigate the relevance of the SI model to human illness and behaviour.

Differential expression analysis detected hundreds of dysregulated genes with approximately three times the number of DEGs in female (n = 1,215) compared to male (n = 400) hippocampal tissue. This is consistent with our behavioural data in these animals [10], which showed females to be more susceptible to SI-induced behavioural measures of anxiety and sociability. GO analysis identified that DEGs in both males and females were enriched in structural and functional synaptic ontologies with DEGs from females implicated in both overall *presynapse* and *postsynapse* ontologies, while DEGs from males were only implicated *postsynapse* ontologies. In the context of our behavioural data [10], this suggests that presynaptic processes may play a role in facilitating SI-induced behaviours of anxiety and decreased sociability observed in the females. Synaptic biology is implicated in many neurodevelopmental disorders. In SCZ, the latest GWAS implicates genes involved in the organisation, differentiation and function of synapses [44]. For ASD, rare variants in genes involved in synapse structure including from the *SHANK* [67], *NRXN* [68] and *NLGN* [69] gene families are linked with the disorder.

In our rare variant analysis, enrichment for genes containing loss-of-function mutations contributing to ASD were found in the DEGs from males but not females. DEGs from both males and females were enriched for genes harbouring DNMs contributing to DD, however the DEGs from males showed a greater level of enrichment. Our sLDSC analysis detected enrichments in SNP heritability associated with SCZ, BPD and IQ in both male and female DEG sets with again males displaying stronger enrichments in all of these phenotypes. Together, the rare variant and heritability analyses indicate that the set of DEGs from male SI

mice, despite only totalling a third of the number of DEGs in the set from female SI mice, converge more strongly with the genes that underpin risk for neurodevelopmental disorders and associated phenotypes in humans.

Anxiety was the most prominent phenotype induced by SI in our behavioural work [10]. Although a nominal enrichment in heritability was seen in the Anx-Ten phenotype for the DEGs from females (p = 0.042), this result did not survive multiple-testing correction. However, it is important to note that anxiety disorders are under less genetic influence (heritability of 20–60%) [70] than for example SCZ and BPD (both with heritability of 60–80% heritable) [44, 47]. As a result and despite a very large sample size, the GWAS of Anx-Ten identified just fourteen independent loci [49] and therefore we were unlikely to detect that our sets of DEGs were strongly enriched for common variants associated with this phenotype. Based on the pretext that no SI-induced cognitive changes were detected in these mice [10], the enrichment in IQ and EA heritability in the set of DEGs from female mice suggests that a subclinical effect of SI on cognition may have been present in this circumstance.

Ten genes were prioritised based on being consistently differentially expressed in males and females, as well as having genome-wide significant associations with any of the phenotypes that were enriched in the sLDSC analysis. Three genes, *RIMS1*, *ZNF365* and *AGAP1*, were associated with two of the four phenotypes tested. RIMS1 plays an important role regulating and localising calcium channels and neurotransmission [71–75], regulating synaptic plasticity [76–79] as well as being crucial for learning and memory in mouse [80]. In humans, *RIMS1* is also an ASD candidate risk gene implicated in rare variant studies [81–83]. *ZNF365* (also called *DBZ*) is involved in neurogenesis, especially regarding basket cells in the somatosensory cortex [84]. It is also involved in oligodendrocyte differentiation [85] and regulating dendritic spine density in pyramidal neurons [86]. *ZNF365* interacts with the SCZ candidate gene, *DISC1*, which through its role on oligodendrocyte differentiation could be a contributor to SCZ and MDD [87]. *AGAP1* (also called *CENTG2*) regulates dendritic spine morphology [88] and is involved in neurotransmitter release in dopaminergic neurons [89]. In humans, variation in AGAP1 has been associated with to ASD [90], SCZ [91]and ASD/ID [92]. Further human data from PsychEncode is consistent with these prioritised mouse DEGs, with RIMS1 downregulated in post-mortem brain samples SCZ (Log2FC = -0.05, FDR = 0.004) and ZNF365 downregulated in post-mortem brain samples of ASD (Log2FC = -0.24, FDR = 0.002) [93].

Although GO analysis primarily implicated changes in synaptic biology caused by SI, the primary findings from cell type enrichment analysis were glial cells. Given the number of cell types analysed here, there is a strict Bonferroni threshold for cell types to be enriched (104 x 2 independent tests). Only cell types with a number highly unique genes overrepresented in our analysis will show up an enriched, which may explain the lack of neuronal cell types showing to be significantly enriched after correction. However, it is now well established that there is a non-neuronal component of neurodevelopment [94]. Astrocytes, enriched in this analysis, play a number of crucial functions to maintain a normal early developmental trajectory. These include maintaining a balanced extracellular environment, protecting neurons during neuroinflammation, and promoting synaptogenesis [95]. In the context of these results, altered expression of genes crucial to astrocyte function may impair the brain's ability to deal with other biological effects of isolation, including neuroinflammation [96] and increased oxidative stress [97]. Human data also highlights the importance of astrocyte function. As discussed in Kruyer, Kalivas [98], human post-mortem data widely implicate morphological and molecular changes in multiple psychiatric disorders including SCZ, MDD and BPD discussed here.

Recent work in the SI field further summarises the effects of SI on the brain [4]. The authors reported overall changes in neuron biology caused by SI in rodents, including changes in neurogenesis, synaptogenesis, neurotransmission and cell morphology. Between GO results

highlighting synaptic ontologies and prioritised genes being strongly implicated in neurotransmitter release, neurogenesis and cell neuron morphology, our findings generally align with these conclusions. Furthermore, there was an emphasis placed on the importance of glial cell types mediating the effects of SI [4]. This is also generally supported by our findings in two ways. First, the prioritised *ZNF365* gene (downregulated in both males and females) plays a role in oligodendrocyte differentiation, which as discussed [4] is a process clearly playing an important role in dysfunction in an SI context. Second, we found that genes are dysregulated as a consequence of SI in hippocampal astrocytes, which as noted [4] are significantly activated as a result of early life SI in females [99].

There are a number of limitations to this present study. First, focusing on a single region limits the ability to generalise these results. Second, these data are generated from a stressor at a single timepoint. Investigating a number of developmental timepoints could help detect transient gene expression changes and could facilitate investigation into the impacts of earlier or later social stressors on gene expression in the mouse brain. Finally, we use a single technique to measure the effect of SI. Other techniques, especially epigenetic profiling using ATAC-seq, or the use of single-cell RNA-seq methods could add valuable context to gene expression profiles found in this data. Addressing these limitations in future studies could help shed light on the relationship between gene expression changes, altered behavioural phenotypes and known human risk variation for neurodevelopmental disorders.

In summary, we provide novel insight into how SI affects the mouse brain. The use of RNA-seq highlights gene expression changes related to synaptic biology that may underpin the changes in behaviour previously found in these animals. We show convergence of DEGs in socially isolated mice with genes containing genetic variation contributing to neurodevelopmental phenotypes in humans. We highlight genes such as *RIMS1*, *ZNF365* and *AGAP1* as candidates for studying disrupted developmental trajectories in disorders such as SCZ, BPD and ASD. Further, these data provide support for the molecular validity of the SI mouse model to study neurodevelopmental phenotypes with further behavioural, molecular and interventional investigations.

## Supporting information

**S1 Table. Differentially-expressed genes (DEGs) detected at FDR < 0.05.**
(XLSX)

**S2 Table. Full Gene Ontology (GO) results—All Ontologies (ConsensusPathDB).**
(XLSX)

**S3 Table. Full Gene Ontology (GO) results—Synaptic Ontologies (SynGO).**
(XLSX)

**S4 Table. Stratified Linkage Disequilibrium Score Regression (sLDSC) full results.**
(XLSX)

**S5 Table. MAGMA genome-wide significant genes.**
(XLSX)

**S6 Table. Full denovolyzeR results (competitive).**
(XLSX)

**S7 Table. Significantly-enriched hippocampal cell types in male and female DEGs.**
(XLSX)

## Author Contributions

**Conceptualization:** John P. Kelly, Gary Donohoe, Derek W. Morris.

**Data curation:** Aodán Laighneach.

**Formal analysis:** Aodán Laighneach.

**Funding acquisition:** Gary Donohoe, Derek W. Morris.

**Investigation:** Aodán Laighneach, John P. Kelly, Lieve Desbonnet, Laurena Holleran, Daniel M. Kerr, Declan McKernan.

**Methodology:** Aodán Laighneach, Lieve Desbonnet, Laurena Holleran, Daniel M. Kerr, Declan McKernan.

**Project administration:** Laurena Holleran, Daniel M. Kerr, Gary Donohoe, Derek W. Morris.

**Supervision:** John P. Kelly, Declan McKernan, Gary Donohoe, Derek W. Morris.

**Writing – original draft:** Aodán Laighneach.

**Writing – review & editing:** John P. Kelly, Declan McKernan, Gary Donohoe, Derek W. Morris.

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
