## [Decision Letter · Decision Letter 0]

19 Oct 2023

PONE-D-23-29486Social isolation-induced transcriptomic changes in mouse hippocampus impact the synapse and show convergence with human genetic risk for neurodevelopmental phenotypesPLOS ONE

Dear Dr. Morris,

Thank you for submitting your manuscript to PLOS ONE. After careful consideration, we feel that it has merit but does not fully meet PLOS ONE’s publication criteria as it currently stands. Therefore, we invite you to submit a revised version of the manuscript that addresses the points raised during the review process.

We look forward to receiving your revised manuscript.

Kind regards,

Alexandra Kavushansky, PhD

Academic Editor

PLOS ONE

4. We note that Figure 1 in your submission contain copyrighted images. All PLOS content is published under the Creative Commons Attribution License (CC BY 4.0), which means that the manuscript, images, and Supporting Information files will be freely available online, and any third party is permitted to access, download, copy, distribute, and use these materials in any way, even commercially, with proper attribution. For more information, see our copyright guidelines: http://journals.plos.org/plosone/s/licenses-and-copyright.

Reviewers' comments:

Reviewer's Responses to Questions

**Comments to the Author**

1. Is the manuscript technically sound, and do the data support the conclusions?

Reviewer #1: Yes

2. Has the statistical analysis been performed appropriately and rigorously? 

Reviewer #1: Yes

3. Have the authors made all data underlying the findings in their manuscript fully available?

Reviewer #1: Yes

4. Is the manuscript presented in an intelligible fashion and written in standard English?

Reviewer #1: Yes

5. Review Comments to the Author

Reviewer #1: This manuscript demonstrates how Post-weaning social isolation (SI) affects the expression of genes in mouse hippocampus and to investigate how these changes related to the genetic basis of neurodevelopmental phenotypes.

The final conclusion is drawn with "a convergence between genes dysregulated by　the SI stressor in the mouse and genes associated with　neurodevelopmental disorders and cognitive phenotypes in humans."

The study appropriately constructs hypotheses based on previous reports and carefully verifies them through experiments. The results of each experiment are appropriately discussed. The manuscript itself is logically and precisely written. Therefore, this paper is basically judged to be worthy of acceptance in this Journal. However, the following items should be reviewed before final decision.

The authors have reported in a previous study that mice exposed to post-weaning SI exhibited a significant increase in anxiety related behaviors (males and females) and decreased sociability (females) compared to animals that were group-housed. I would like to confirm whether behavioral disorders as a phenotype are also expressed under the present conditions.

In addition, please include a discussion of the possibility that any changes at the genetic level in animals subjected to post-weaning SI will always lead to behavioral disorders as a phenotype. In terms of genetic changes and phenotypic expression, is it possible to indicate specifically to what extent they are correlated? If so, please indicate it.

6. PLOS authors have the option to publish the peer review history of their article (what does this mean?). If published, this will include your full peer review and any attached files.

Reviewer #1: No

---

## [Author Response · Author response to Decision Letter 0]

27 Oct 2023

Responses to Reviewer’s Comments:

1. This manuscript demonstrates how Post-weaning social isolation (SI) affects the expression of genes in mouse hippocampus and to investigate how these changes related to the genetic basis of neurodevelopmental phenotypes.

The final conclusion is drawn with "a convergence between genes dysregulated by the SI stressor in the mouse and genes associated with　neurodevelopmental disorders and cognitive phenotypes in humans."

The study appropriately constructs hypotheses based on previous reports and carefully verifies them through experiments. The results of each experiment are appropriately discussed. The manuscript itself is logically and precisely written. Therefore, this paper is basically judged to be worthy of acceptance in this Journal. However, the following items should be reviewed before final decision.

Response: We thank the reviewer for the positive comments regarding the study design, writing and discussion.

2. The authors have reported in a previous study that mice exposed to post-weaning SI exhibited a significant increase in anxiety related behaviors (males and females) and decreased sociability (females) compared to animals that were group-housed. I would like to confirm whether behavioral disorders as a phenotype are also expressed under the present conditions.

Response: The tissue used in this study is from a subgroup of the animals used in Desbonnet et al. (2022) in which the behavioural and immunological aspects of these animals were studied. This present study, which begun much later than the original study, assays gene expression in the hippocampus of the same socially-isolated and group-housed adult mice. The behavioural changes, summarised in Table 1 of Desbonnet et al. (2022) apply to these animals. Therefore, we can confirm the anxiety (males and females) and decreased sociability (females) phenotypes were expressed under the present conditions. The Materials and Methods states: “Tissues used in this analysis were obtained directly from the animals used in our previous study [Desbonnet et al. (2022)].”

3. In addition, please include a discussion of the possibility that any changes at the genetic level in animals subjected to post-weaning SI will always lead to behavioral disorders as a phenotype. In terms of genetic changes and phenotypic expression, is it possible to indicate specifically to what extent they are correlated? If so, please indicate it.

Response: This is an important point. First, we must consider the limitations of using a single timepoint (post-weaning), analyzing a single region (hippocampus) and using a single functional genomics technique (RNA-seq) in comparing genetic data and behavioural data. Although the hippocampus was chosen based on its established roles in stress response and emotion (Cameron and Glover 2015, Snyder et al. 2011), we may have missed out on behaviourally relevant region-specific expression changes in other parts of the mouse brain. Furthermore, by only using bulk RNA-seq, we may be missing out on important cell-type specific and epigenetic context to our data which could help more precisely link genetic changes and behaviour. 

Regarding correlations, it is difficult to assess the true relationship between gene expression and behavioural changes in mouse models. Published studies will generally have a higher concordance of behavioural and molecular data. For example, of the targeted studies showing anxiety mentioned in the introduction section of this manuscript excluding Desbonnet et al. (2022), 6 out of 7 (86%) report on protein or gene expression and all of them show changes. Although it is certainly possible for gene expression changes to be present in the absence of significant behavioural changes e.g., an underpowered behavioural analysis, it’s highly unlikely that behavioural changes will be present in the absence of gene expression changes. Therefore, we generally consider behavioural changes a fundamental component to studies of social stress as it represents the strongest evidence of true response to the stressor. 

Specific to our data, we are looking at global changes at a single timepoint without particular focus on individual genes. Therefore, a direct correlation between behavioural phenotypes and gene expression would be impractical. However, based on this comment, we could consider future more targeted studies (i.e. using lower throughput techniques such as qPCR) which focus on the specific relationship between gene expression and behaviour phenotype magnitudes. Furthermore, the incorporation of various developmental timepoints may further help understand and correlate these relationships. 

Actions taken: We have added a paragraph in the discussion section, highlighting the limitations of this work. See below:

“There are a number of limitations to this present study. First, focusing on a single region limits the ability to generalise these results. Second, these data are generated from a stressor at a single timepoint. Investigating a number of developmental timepoints could help detect transient gene expression changes and could facilitate investigation into the impacts of earlier or later social stressors on gene expression in the mouse brain. Finally, we use a single technique to measure the effect of SI. Other techniques, especially epigenetic profiling using ATAC-seq, or the use of single-cell RNA-seq methods could add valuable context to gene expression profiles found in this data. Addressing these limitations in future studies could help shed light on the relationship between gene expression changes, altered behavioural phenotypes and known human risk variation for neurodevelopmental disorders.”

Cameron, H. A. and Glover, L. R. (2015) Adult neurogenesis: beyond learning and memory. Annu Rev Psychol, 66, pp. 53-81.

Desbonnet, L., Konkoth, A., Laighneach, A., McKernan, D., Holleran, L., McDonald, C., Morris, D. W., Donohoe, G. and Kelly, J. (2022) Dual hit mouse model to examine the long-term effects of maternal immune activation and post-weaning social isolation on schizophrenia endophenotypes. Behav Brain Res, 430, pp. 113930.

Koopmans, F., van Nierop, P., Andres-Alonso, M., Byrnes, A., Cijsouw, T., Coba, M. P., Cornelisse, L. N., Farrell, R. J., Goldschmidt, H. L., Howrigan, D. P., Hussain, N. K., Imig, C., de Jong, A. P. H., Jung, H., Kohansalnodehi, M., Kramarz, B., Lipstein, N., Lovering, R. C., MacGillavry, H., Mariano, V., Mi, H., Ninov, M., Osumi-Sutherland, D., Pielot, R., Smalla, K. H., Tang, H., Tashman, K., Toonen, R. F. G., Verpelli, C., Reig-Viader, R., Watanabe, K., van Weering, J., Achsel, T., Ashrafi, G., Asi, N., Brown, T. C., De Camilli, P., Feuermann, M., Foulger, R. E., Gaudet, P., Joglekar, A., Kanellopoulos, A., Malenka, R., Nicoll, R. A., Pulido, C., de Juan-Sanz, J., Sheng, M., Sudhof, T. C., Tilgner, H. U., Bagni, C., Bayes, A., Biederer, T., Brose, N., Chua, J. J. E., Dieterich, D. C., Gundelfinger, E. D., Hoogenraad, C., Huganir, R. L., Jahn, R., Kaeser, P. S., Kim, E., Kreutz, M. R., McPherson, P. S., Neale, B. M., O'Connor, V., Posthuma, D., Ryan, T. A., Sala, C., Feng, G., Hyman, S. E., Thomas, P. D., Smit, A. B. and Verhage, M. (2019) SynGO: An Evidence-Based, Expert-Curated Knowledge Base for the Synapse. Neuron, 103(2), pp. 217-234 e4.

Snyder, J. S., Soumier, A., Brewer, M., Pickel, J. and Cameron, H. A. (2011) Adult hippocampal neurogenesis buffers stress responses and depressive behaviour. Nature, 476(7361), pp. 458-61.

---

## [Decision Letter · Decision Letter 1]

30 Nov 2023

Social isolation-induced transcriptomic changes in mouse hippocampus impact the synapse and show convergence with human genetic risk for neurodevelopmental phenotypes

PONE-D-23-29486R1

Dear Dr. Morris,

We’re pleased to inform you that your manuscript has been judged scientifically suitable for publication and will be formally accepted for publication once it meets all outstanding technical requirements.

Kind regards,

Alexandra Kavushansky, PhD

Academic Editor

PLOS ONE

Additional Editor Comments (optional):

Reviewers' comments:

Reviewer's Responses to Questions

**Comments to the Author**

1. If the authors have adequately addressed your comments raised in a previous round of review and you feel that this manuscript is now acceptable for publication, you may indicate that here to bypass the “Comments to the Author” section, enter your conflict of interest statement in the “Confidential to Editor” section, and submit your "Accept" recommendation.

Reviewer #1: All comments have been addressed

2. Is the manuscript technically sound, and do the data support the conclusions?

Reviewer #1: (No Response)

3. Has the statistical analysis been performed appropriately and rigorously? 

Reviewer #1: (No Response)

4. Have the authors made all data underlying the findings in their manuscript fully available?

Reviewer #1: (No Response)

5. Is the manuscript presented in an intelligible fashion and written in standard English?

Reviewer #1: (No Response)

6. Review Comments to the Author

Reviewer #1: In this revised manuscript, the authors have appropriately addressed the reviewers' comments, and the paper is considered acceptable to the Journal at this stage.

7. PLOS authors have the option to publish the peer review history of their article (what does this mean?). If published, this will include your full peer review and any attached files.

Reviewer #1: No

---

## [Editor Report · Acceptance letter]

12 Dec 2023

PONE-D-23-29486R1 

PLOS ONE

Dear Dr. Morris, 

I'm pleased to inform you that your manuscript has been deemed suitable for publication in PLOS ONE. Congratulations! Your manuscript is now being handed over to our production team.

Kind regards, 

on behalf of

Dr. Alexandra Kavushansky 

Academic Editor

PLOS ONE